# Prevalence of Comorbidity and Its Effects on Sickness-Absenteeism among Brazilian Legislative Civil Servants

**DOI:** 10.3390/ijerph20065036

**Published:** 2023-03-13

**Authors:** Francisco Edison Sampaio, Manuel Joaquim Silva Oliveira, João Areosa, Emílio Facas

**Affiliations:** 1Faculty of Engineering, University of Porto (FEUP), 4200-465 Porto, Portugal; 2Higher School of Business Sciences, Polytechnic Institute of Setubal, 2914-503 Setubal, Portugal; 3Department of Social and Work Psychology, University of Brasilia (UNB), Brasília 70910-900, Brazil

**Keywords:** chronic disease, comorbidity, multimorbidity, sickness absenteeism, comorbidity index, civil servant

## Abstract

Studies have shown there is an association of chronic diseases with working days lost, considering the impact of these pathologies on the levels of vulnerability of the individual’s health, with an increased risk of work disability. This article is part of a more comprehensive investigation on the sickness absenteeism of civil servants of the legislative branch in Brazil, with the purpose of determining the comorbidity index (CI) of the individuals and its correlation with days of absence from work. Sickness absenteeism was counted from the data of 37,690 medical leaves, from 2016 to 2109, involving 4149 civil servants. The self-administered comorbidity questionnaire (SCQ) was used to estimate the CI, based on the diseases or chronic health problems declared by the participants. The average number of working days lost per servant per year was 8.73 days, totaling 144,902 days of absence. The majority of the servants (65.5%) declared at least one chronic health condition. A significant association between the CI scores and working days lost was observed (r = 0.254, *p*-value < 0.01), thus showing that the CI may be an important predictor of sickness absenteeism. Chronic diseases or health problems are a characteristic of the general population, often affecting working capacity.

## 1. Introduction

Work absenteeism is a global phenomenon that affects public and private organizations with detrimental effects on employers, employees, government, and society. Globally, annual statistics indicate the occurrence of about 374 million non-fatal work accidents, which cause at least four days of absence from work. In addition, over 2.7 million worker deaths from occupational accidents or illnesses have been reported [1]. In Europe, sickness absenteeism averaged 11.5 days per worker per year from 1970 to 2019 [2]. In Brazil, a total of 2,934,155 occupational accidents were recorded from 2015 to 2019, of which 34% caused at least 16 days away from work. Of these accidents or occupational diseases, 85,603 occurrences involved Brazilian civil servants from the executive, legislative, and judiciary branches [3]. Moreover, the International Labor Organization (ILO) has reported negative economic effects at around 3.94% of the Gross Domestic Product worldwide each year due to work absenteeism [1].

Several studies on sickness absenteeism in the public service have reported some pathological predictors of work absences, including mental health problems, musculoskeletal disorders, trauma to different parts of the body, and respiratory system diseases, among others [4,5,6,7]. In this context, individuals with comorbidity may present a higher degree of absence from work due to the level of human vulnerability. In Australia, individuals with comorbidity reported greater symptom severity, poorer work performance, and a greater number of working days lost when compared to individuals with more favorable health conditions [8].

### 1.1. Work Absenteeism and Sickness Absenteeism

Work absenteeism is the absence or non-attendance of the worker at work, regardless of the reason and legality [9]. As reported by De Oliveira (2016, p. 14) [10], absenteeism can be understood as a phenomenon of multifactorial etiology, characterized by an unplanned absence from the workplace. In addition, absenteeism is understood as absences due to personal health problems, leading to medical leaves [4]. According to the ILO, absenteeism is the absence from work due to a worker’s incapacity for the working tasks, caused by illness or an accident injury, or risk of transmission of some disease, excluding non-attendance at work resulting from pregnancy and imprisonment [11]. Sickness absenteeism is a phenomenon that affects private and public organizations since all work exercises can expose the worker to dangers inherent to the environment and general working conditions, regardless of the type of work. As reported by Rodrigues et al., (2013, p. 138) [5], absenteeism is a worldwide phenomenon of multidimensional character, resulting from non-specific complaints and declared illnesses, and its occurrence and evolution are influenced by socio-demographic, behavioral, and occupational factors, among others.

### 1.2. Comorbidity

The term comorbidity does not reflect a conceptual consensus or convergence regarding its application in health care, with distinct approaches from several authors and health institutions. However, there is no disagreement about the association of this health condition with incapacity to work, and therefore absenteeism from work. Briefly, comorbidity has been described as the coexistence of two or more diseases in a patient, taking as reference an index disease, while multicomorbidity comprises an equivalent condition regarding the presence of diseases or chronic health problems in the same individual, without considering a medical condition as reference or dominant [7,8]. When dealing with the conceptualization of comorbidity, Valderas et al., (2009) [12] recognize that although there is no agreement on the subject, comorbidity is most often defined according to a specific index condition (main or reference disease). In this regard, the author presents Feinstein’s definition, which describes comorbidity as any additional different entity that has existed or may occur during the clinical course of a patient with an index disease under study. On the other hand, the author brings the concept of multimorbidity as the “co-occurrence” of multiple chronic or acute diseases and medical conditions in the same individual, with no reference to an index condition. As the proponents of the term multimorbidity prefer to focus on primary care, the term index disease is often not applied. Valderas et al., (2009) [12] refer to the morbidity burden, defined as the total burden of physiological dysfunction, or diseases with some impact on the physiological reserve of an individual. In turn, the term patient complexity can be defined as an interaction between the socioeconomic, cultural, environmental, and behavioral characteristics and the health conditions of the individuals, which can exert an influence on the morbidity burden. Figure 1 shows a graphic representation of these concepts.

According to the authors, comorbidity and multimorbidity are associated with adverse health outcomes. Eventually, the emphasis on an index disease may be important in specialized care. On the other hand, when primary care is the interest, the burden of multimorbidity becomes the focus, thus the patient should be treated as a whole, without privileging any specific medical condition [13]. Depending on the perspective of analysis, the differentiation between multimorbidity and comorbidity can be unreal since the same individual can be considered in both situations [13]. The conceptual relevance of comorbidity and multimorbidity is due to the impact of the approach on healthcare systems dealing with patients with multiple chronic conditions, which leads to a direction in research. In this regard, the term “comorbidity” was first proposed in 1970 by Feinstein to describe any pathology or health problem additional to an index disease. However, since 1976, the term “multimorbidity” has been used more frequently by researchers to designate the same thing as morbidity [14]. Due to a growing ambiguity in the adoption of the two terms, in 1996 Van den Akker et al. suggested keeping Feinstein’s original concept for comorbidity, while multimorbidity was defined as the occurrence of multiple chronic or acute diseases and medical conditions in the same individual [14].

Due to their representativeness in individual health, comorbidity and multicomorbidity should be considered highly relevant variables in studies aimed to estimate absenteeism due to illness in organizations, whether private or public. This article is part of a larger investigation on sickness absenteeism of civil servants that work in the Legislative Houses (Federal Senate, House of Representatives, and Legislative Assembly of the State of Goias—ALEGO) in Brazil, which aims to evaluate the prevalence of diseases or chronic health problems of the participants, as well as the effects of this medical condition on the number of working days lost by these servants.

## 2. Materials and Methods

### 2.1. Experimental Design

The present study consists of an observational, cross-sectional, and analytical quantitative approach to sickness absenteeism of civil servants working in the legislative branch in Brazil. The research was developed with the approval of the Ethics Committee of the Federal University of Goias, under register number 3.962.630 on 9 April 2020.

### 2.2. Participants

The counting of sick leave involved a total of 4149 servants who were absent from work for health reasons between January 2016 and December 2019. To evaluate the individual health condition determined through the comorbidity index, 447 electronic questionnaires answered voluntarily by the participants were validated, which meets the sample sizing requirements for a sampling error ≤ 0.5 and confidence level = 95%.

### 2.3. Measurements and Data Collection

The medical leaves with the reasons for absence and the respective working days lost by the servants, as well as the sociodemographic and occupational information, were made available by the organizations participating in the research, in a spreadsheet developed especially for this purpose. The health condition of the servants was evaluated using the self-administered comorbidity questionnaire (SCQ), developed, and validated by Sangha et al., (2003) [15]. All absent servants who agreed to participate in the study answered the SCQ in electronic format, under the condition of anonymity, which allowed for estimating the CI of the participants. This instrument allows the evaluation of the comorbidity condition from individual responses about the presence of diseases/chronic health problems, necessary medical treatments, and limitations imposed by medical conditions in the execution of activities. According to the SCQ, the higher the incidence of chronic diseases, combined with the need for medical treatment and restrictions in performing activities, the more serious the state of health. People with high CIs may have a higher level of personal health vulnerability, leading to a greater likelihood of work absences, especially when their health status is ignored in the workplace. Concerning the SCQ score, an individual can score a maximum of 3 points for each medical condition, consisting of 1 point for the presence of the active health problem, 1 point for the existence of medical treatment, and an additional point in case of functional limitation. The questionnaire presents 13 health problems and 3 additional possibilities, totaling a maximum score of 48 points or 39 points when the open items or closed items are used, respectively. Comorbidity was expressed as an index (CI) with a value between 0.00 (no morbidity) and 1.00 (maximum score on the questionnaire), obtained by the ratio between the score achieved by each individual and the maximum possible score.

### 2.4. Statistical Analysis

The counting and preliminary treatment of the sick leave data as well as the CI calculation were performed using Microsoft Office’s Excel software, version 2302. The statistical analysis of absolute and relative frequencies and measures of position and dispersion and the correlation between variables were performed using IBM^®^ SPSS Statistics (Statistical Package for the Social Sciences, Inc., Chicago, IL, USA)

## 3. Results

### 3.1. Sociodemographic and Occupation Characteristics

The servants that responded to the comorbidity survey (n = 447) were predominantly male (52.7%), married/stable union (69.1%), and a mean age above 46 years. Table 1 shows the sociodemographic details of the participants.

### 3.2. Global Sickness Absenteeism Data

The work absences of civil servants from the three Legislative Houses (LH) were counted by the number of medical leaves (ML) granted and working days lost, which was taken as the parameter to portray absenteeism due to illness. In the study period (2016 to 2019), LHs issued a total of 37,690 ML, involving 4149 servants, which resulted in 144,902 working days lost. Table 2 presents a summary of these absences.

As shown in Table 2, The number of working days lost is quite different among the three LHs, probably due to the number of servants in these Legislative Houses and the criteria used by these organizations to measure absenteeism. The House of Deputies and the Federal Senate issue ML for absences starting from 1 day of absence due to health problems, while ALEGO recorded only absences of more than 3 days. The other cases of non-attendance at work, less than 4 days, were managed by the immediate superior. It is worth mentioning the annual average of working days lost per server (d/s/yr), as shown in the last column of Table 2, as it represents the real average of sickness absenteeism. In this sense, ALEGO and the House of Deputies had a very similar performance, around 10 d/s/yr, while the Federal Senate positioned well above these figures, with an average absence of 6.2 d/s/yr. It should be noted that, except for very few cases, ALEGO did not compute absences of 1 to 3 days in its general absenteeism register, which may have affected its average work absences. On the other hand, the much more favorable situation of the Federal Senate may be associated with a more adequate general working condition, among other factors.

### 3.3. Individual Health Condition (Comorbidities)

The individual health status of the servants regarding the presence of active or chronic health problems was evaluated using an electronic questionnaire (self-administered comorbidity questionnaire—SCQ), which allows the calculation of the CI. This index allows for estimating the situation of individuals regarding the existence or absence of permanent or long-term morbidities. An individual who declares no pathology or health problem has a CI score of 0.00. An individual with a pathology or chronic health problem without the need for medical treatment or restrictions in the performance of any type of activity receives a CI score of 0.02. Table 3 shows the CI scores determined in the three LHs.

A total of 154 servants (34.5%) declared no chronic health problems. In contrast, most respondents (65.5%) reported at least one chronic problem/illness. The highest frequency score (CI = 0.04) was registered in 77 cases and may correspond to the presence of two comorbidities or only one morbidity combined with the need for medical treatment, or difficulties to perform activities. The most serious situation, CI = 0.44, was declared by only one servant and corresponds to a health condition that can show a significant vulnerability. However, all cases with CI scores above 0.20 deserve more attention because it implies the presence of at least 4 pathologies/health problems combined with medical treatment and difficulties to perform activities.

Table 4 presents the distribution of individuals according to the number of diseases or health problems. Among those who declared a diagnosis of chronic diseases (293 individuals), 44.4% reported a single occurrence, while the remaining individuals, 55.6%, reported having two or more comorbidities.

Among the items suggested in the SCQ and those included by the participants, 622 records of diseases were reported, 362 referring to those in the questionnaire, and 260 new items informed by the servants. Among the chronic health problems included in the questionnaire, back pain was the most frequent, with 126 records, followed by hypertension and depression, with 70 and 54 records, respectively. On the other hand, the pathologies/chronic health problems directly declared by the respondents as “Other Health Problems”, in the categories of Problem-1, Problem-2, and Problem-3, resulted in 156, 74, and 30 records, respectively. Figure 2 presents a graph with the frequency of the diseases/health problems, in percentages, resulting from the CI estimation among the servants from all LHs. All diseases/health problems reported by the participants (differing from those in the questionnaire) are represented by the categories Other Health Problems 1, 2, and 3, considering the vast list of specific pathologies reported by the servants. Figure 3 shows an overview of self-reported chronic diseases, in the form of a word cloud, highlighting back pain, hypertension, depression, and other musculoskeletal disorders (Other-DME) with higher frequencies.

Other pathologies and health problems reported by the servants also proved to be very important in the estimation of the CI score. In order, the cases of diabetes, ulcers, anxiety, and hypothyroidism stand out. It is also worth noting the cases of cancer and kidney disease.

The CI score becomes more significant when it also reflects the need for the respective medical treatment, which denotes a condition that requires permanent control. Moreover, the health condition can evolve and become more serious when the comorbidity situation prevents or makes it difficult for individuals to perform work activities. Figure 4 and Figure 5, respectively, show the percentage of servants with chronic diseases/health problems requiring medical treatment and those who have difficulties in performing work activities.

As shown in Figure 4, the vast majority of the civil servants with CI scores above zero (83%) received some type of treatment for their health situation. Moreover, 54% of these servants with comorbidities reported no difficulty performing any activity given their health condition. In contrast, 46% informed that the presence of chronic pathologies/health problems causes restrictions in performing activities.

The presence of chronic diseases or health problems can make individuals more vulnerable due to several factors, including a risk to personal health, and greater susceptibility of these individuals, due to daily exposures, to risk factors (at work or outside work), which can increase the degree of severity of the effects resulting from exposure.

### 3.4. Correlation between Comorbidity Index (CI), Working Days Lost, and Sociodemographic Variables

The association between the individual health status measured by the CI, and the absenteeism measured in working days lost, as well as the sociodemographic variables were analyzed through bivariate correlation. Higher CI scores indicate a level of increased vulnerability to individuals’ health and may influence work absences. Table 5 shows the result of this analysis through Pearson’s coefficient.

The CI scores were positively and significantly associated with the work absences of the servants at a 1% level (r = 0.254, *p*-value < 0.01), thus an increase in the level of vulnerability of the individual’s health due to chronic diseases is associated with an increase in the working days lost. Moreover, the CI was positively and significantly associated with both the age of the servants at a 5% level (r = 0.116, *p*-value < 0.05) and the length of service (seniority) at a 1% level (r = 0.133, *p*-value < 0.01). Therefore, an increase in the age of the servants and their length of service is correlated with an increase in the CI score, which can lead to a longer time away from work. It should also be noted that age and seniority were statistically highly significant at a 1% level (r = 0.762, *p*-value < 0.01). Thus, the greater length of time on the job implies an increase in the servants’ age, which represents a favorable situation for higher CI scores.

Sickness absenteeism, as already mentioned, is a multifactorial phenomenon; that is, it consists of a variable dependent on several other variables that act as predictors of work absences motivated by health problems. However, the CI proved to be a possible predictor of relative importance, considering the moderate correlation (r = 0.254) with the working days lost by the legislative servants from all LHs.

Regarding the prevalence of comorbidities according to the gender of the participants, no significant differences between genders were observed through the independent t-test (t 445 = 0.039; *p*-value = 0.969). There may be qualitative differences once male and female servants present differences in terms of the type of diseases or health problems.

## 4. Discussion

### 4.1. Sickness Absenteeism

The average annual rate of days of absence of civil servants of the three Legislative Houses in this study was 8.73 d/s/yr, which is relatively moderate when compared to other national and international cases of work absences in the public service due to health problems. The sickness absenteeism of civil servants of the City Halls of Goiania, between 2005 and 2010, and Coritiba, between 2010 and 2015, had an average of working days lost of 12.07 d/s/yr [4] and 23.04 d/s/yr, respectively [6]. Similar studies with civil servants in Canada and Australia indicated an average civil servant absence of 11.6 d/s/yr [5]. However, much lower values of days away from work in the public sector have been reported, as in the case of the UK countries that achieved an average annual working day loss of 4.4 d/s/yr in 2018.

The poorer results of sick leave in the legislative houses of the present study may be related to the less adverse general working conditions in the executive branch, mainly in the areas of education, health, and public security. These areas present frequent occurrences of absence from work due to health problems, considering the pathological potential of these sectors, due to direct contact with the public, high social demand for such services, or operational difficulties faced by educators and health and public safety professionals. Moreover, the legislative servants participating in this study perform only internal administrative activities of support to parliamentarians and have their own health service that provides care, including basic health promotion actions.

### 4.2. Comorbidity

The presence of chronic health diseases/conditions in the general population is a reality for millions of people. In Brazil, according to the National Health Survey, in 2019, 52% of the population aged 18 years or older reported having been diagnosed with at least one chronic disease, with hypertension standing out with 23.9% of individuals, and depression reaching 10.2% of the population [16]. This situation seems to be more comprehensive with the working population. The results of this study showed that most legislative staff (65.5%) reported at least one chronic disease or health problem, while the others (34.5%) reported no chronic pathology. Among the servants with chronic diseases or health problems, 44.4% reported a single occurrence, while 55.6% reported two or more comorbidities. A study conducted with more than 10,000 workers in the general population in Denmark found that 56.8% of the participants had one or more chronic diseases, which were associated with the risk of leaving for health treatment [17], corroborating the present research.

Back pain, hypertension, and depression were the pathologies with the highest frequency for all LHs, reaching 19.6, 11.0, and 8.8%, respectively, of all reported chronic diseases/problems, considering the category Other Health Problems 1, 2, and 3. Several studies corroborate the prevalence of these pathologies among workers and the general population, including civil servants. Serranheira et al., (2020) [18] investigated 735 workers from different occupational fields and reported that 69% of the respondents presented at least one episode of low back pain in 12 months, with the highest proportion of individuals presenting more than six episodes of low back pain per year among civil servants (31.8%). Research involving 4844 public service workers in Nigeria found a prevalence of 35% of cases of hypertension and 36.4% of prehypertension, with a slight predominance among male employees, while only 2% of employees diagnosed with hypertension were aware of their health condition [19]. Finally, a study evaluated the factors associated with temporary work incapacity among Brazilian university servants and found 30% of recurrent depressive disorders among 1753 cases of temporary incapacity for the 21 most prevalent diseases studied [20].

### 4.3. Comorbidity and Sickness Absenteeism

In general, sickness-related absenteeism is an outcome variable of several other variables within the work environment, in addition to external and individual factors. In this study, the main purpose was to assess the role of chronic diseases or problems in work absences.

The statistical analyses revealed that the CI that contemplates the presence of pathologies/chronic health conditions was highly significantly associated with the working days lost (r = 0.254, *p*-value = 0.01), therefore the number of days away from work increases with the increase in the CI score. This correlation was evidenced in other research involving civil servants. It is known that the chance of absenteeism among workers with chronic diseases is 6.34 times higher when compared to those in the opposite situation. Moreover, there is a higher probability of the occurrence of negative critical incidents with these individuals at work [21]. The presence of chronic diseases associated with a low capacity (physical and mental) for work is correlated with a high risk of long-term absence in the general working population [17]. A study on the association between comorbidities and general labor force participation of Australian workers with back pain showed that an individual with these conditions and heart disease was ten times more likely to be out of the labor force. Absenteeism was also associated with long-term work incapacity among workers with episodes of comorbid depressive disorders or anxiety [22]. Finally, multicomorbidity is common among young adult workers and is related to absenteeism as well as presenteeism at work [23].

In this study, a significant and positive correlation between the CI scores and age and length of service (seniority) was observed, i.e., the level of vulnerability of individual health of legislative workers to diseases and chronic health problems increases with increasing age and length of service. This finding demands attention since in some countries the multimorbidity rates for the population over 65 years of age are estimated at 80–90%, which may represent a greater susceptibility of older workers to the onset of diseases, leading to a greater occurrence of work disability or permanent disability to work.

The results showed no significant differences between men and women regarding the CI scores for the total number of participants, regardless of age or activity. This result may vary when some specific criteria are considered, including education, age, or specific pathology, among others. However, there is usually a prevalence of chronic diseases in the group of women in the general population. Regarding the population aged 18 years or older in Brazil in 2019, hypertension was more prevalent among women (26.4%) when compared to men (21.1%). Likewise, regarding diagnoses of depression, women showed a prevalence of 14.7% when compared to 5.1% among men [16]. More specific population extracts, such as the legislative staff participating in this research, do not necessarily reproduce the general profile of the population in terms of comorbidities.

## 5. Conclusions

The main purpose of this study was to evaluate the chronic health condition among civil servants that work in the legislative houses in Brazil, through the comorbidity index, as well as its effects on sickness-related absenteeism, expressed in working days lost.

The statistical analyses revealed that the vast majority of the participants had at least one chronic disease or health problem. In turn, the comorbidity index showed that at least 8 out of 10 of these individuals use medication or other medical treatment, and no less than four individuals reported difficulties or restrictions in performing some activity due to their health condition. Thus, it is reasonable to conclude that the population under study presents a profile strongly characterized by the presence of chronic health conditions, which affect the personal health of these individuals, imposing the need for some kind of medical monitoring and the risk of losing working capacity.

Regarding the effects of the chronic health condition on work absences assessed by the comorbidity index, it was evident that the individual health condition was strongly associated with the working days lost by the civil servants with diseases or chronic problems. Therefore, the presence of diseases or chronic health problems had important effects on the work absenteeism of this group of civil servants. As already discussed, absenteeism is a phenomenon of multifactorial etiology and may be associated with internal and external factors to work. However, individual characteristics such as the socio-demographic profile and individual health conditions act to mitigate or aggravate absences from work activities.

Studies aimed at detailing the prevalence of chronic health problems of workers and the impact of this medical condition on the reduction of working capacity can help in understanding sickness-related absenteeism, providing a more adequate and humanized management of work, with possible positive effects on people’s health and productivity.

## Figures and Tables

**Figure 1 ijerph-20-05036-f001:**
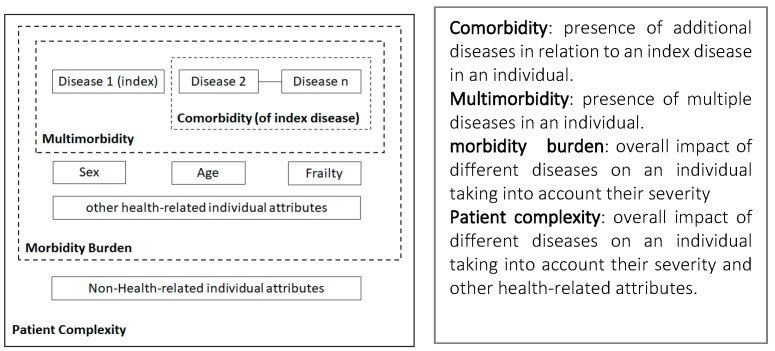
Comorbidity constructs, adapted from (Valderas et al., 2009 [12]).

**Figure 2 ijerph-20-05036-f002:**
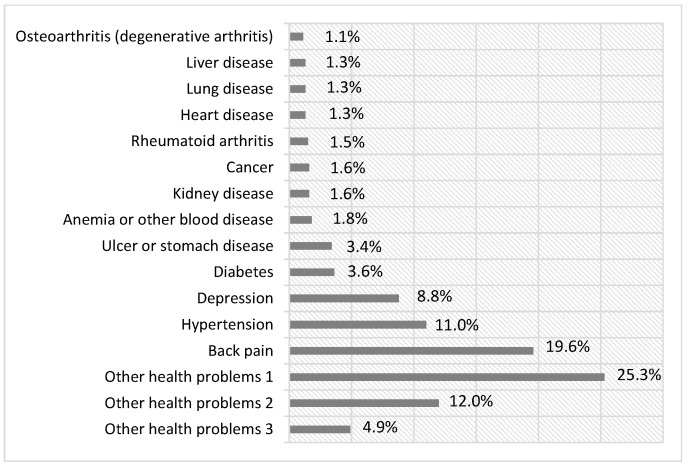
Frequency of chronic diseases/health problems.

**Figure 3 ijerph-20-05036-f003:**
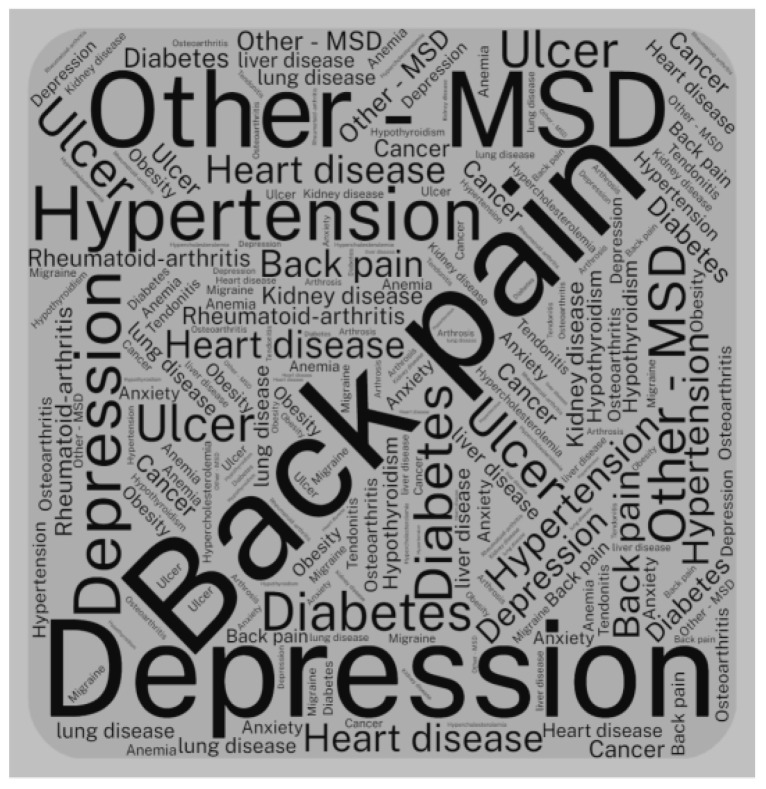
Chronic diseases/health problems among civil servants.

**Figure 4 ijerph-20-05036-f004:**
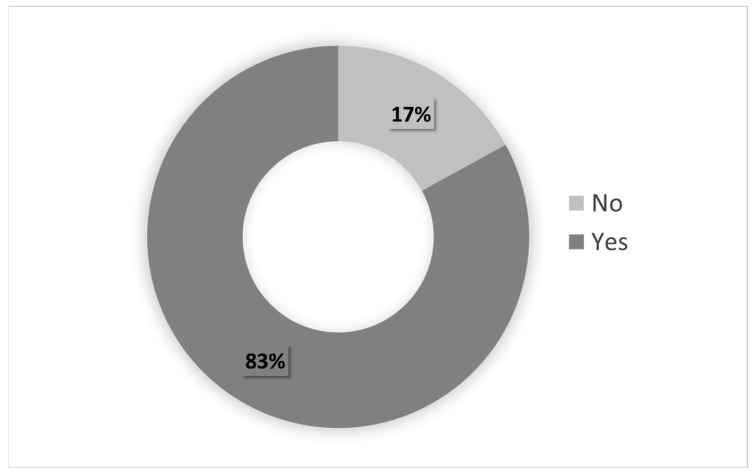
Individuals with CI > 0.00 with medical treatment.

**Figure 5 ijerph-20-05036-f005:**
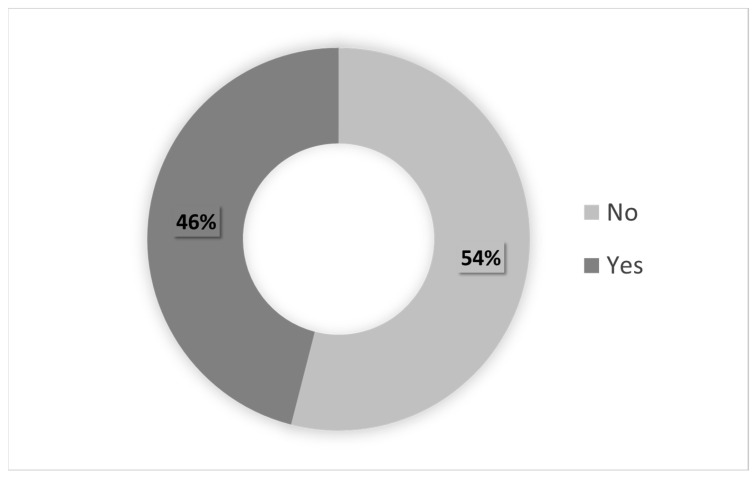
Individuals with CI > 0.00, with and without difficulties to perform activities.

**Table 1 ijerph-20-05036-t001:** Sociodemographic and occupation characteristics of the participants.

Variables		ALEGO	House of Deputies	Federal Senate	All LH
		n (%)	n (%)	n (%)	n (%)
Gender	Male	11 (61.1)	117 (48.4)	104 (57.8)	232 (52.7)
	Female	7 (38.9)	125 (51.6)	76 (42.2)	208 (47.3)
Marital status	Single/widowed	2 (11.1)	37 (15.3)	53 (29.5)	92 (20.9)
	Married/stable union	12 (66.7)	187 (77.3)	105 (58.3)	304 (69.1)
	Separated	4 (22.2)	18 (7.4)	22 (12.2)	44 (10.0)
Education	Elementary school	1 (5.6)	0 (0.0)	1 (0.6)	2 (0.4)
	High school	2 (11.1)	6 (2.5)	4 (2.2)	12 (2.7)
	College	7 (38.9)	29 (12.0)	33 (18.3)	69 (15.7)
	Specialization	6 (33.3)	141 (58.3)	110 (61.1)	257 (58.4)
	Master	2 (11.1)	44 (18.2)	26 (14.5)	72 (16.4)
	PhD	0 (0.0)	22 (9.0)	6 (3.3)	28 (6.4)
Age	M(SD) ^1^	43.1 (10.0)	46.6 (7.6)	45.5 (9.1)	46.3 (9.4)
Working hours	M(SD) ^1^	6.4 (0.6)	8 (0.0)	7.6 (0.6)	7.8 (0.5)
Length of service (Seniority)	M(SD) ^1^	12.4 (12.4)	14.5 (8.3)	14.2 (9.7)	15.8 (10.5)

^1^ Mean (Standard deviation).

**Table 2 ijerph-20-05036-t002:** Sickness absenteeism in the 3 Legislative Houses, from 2016 to 2019.

All Houses	Absences (Days)	Percentage (%)	Number of Absent Servants	Percentage (%)	Mean (d/s/yr)
ALEGO	4272	2.9	107	2.6	9.98
House of Deputies	103,981	71.8	2567	61.9	10.13
Federal Senate	36,649	25.3	1475	35.6	6.21
Total	144,902	100	4149	100	8.73

**Table 3 ijerph-20-05036-t003:** Comorbidity index (CI) for all LHs.

CI	Frequency	Percentage (%)	Cumulative Percentage (%)
0.00	154	34.5	34.5
0.02	29	6.5	40.9
0.04	77	17.2	58.2
0.06	41	9.2	67.3
0.08	33	7.4	74.7
0.10	16	3.6	78.3
0.13	30	6.7	85.0
0.15	18	4.0	89.0
0.17	11	2.5	91.5
0.19	12	2.7	94.2
0.21	6	1.3	95.5
0.23	2	0.4	96.0
0.25	8	1.8	97.8
0.27	2	0.4	98.2
0.29	1	0.2	98.4
0.31	3	0.7	99.1
0.33	2	0.4	99.6
0.35	1	0.2	99.8
0.44	1	0.2	100.0
Total	447	100.0	

**Table 4 ijerph-20-05036-t004:** Distribution of comorbidity events.

Number of Diseases or Health Problems	Individuals	Percentage	Cumulative Percentage
0	154	34.5	34.5
1	130	29.1	63.5
2	65	14.5	78.1
3	52	11.6	89.7
4	33	7.4	97.1
5	7	1.6	98.7
6	4	0.9	99.6
7	1	0.2	99.8
9	1	0.2	100.0
Total	447	100.0	

**Table 5 ijerph-20-05036-t005:** Correlations between CI, working days lost, and sociodemographic variables.

Correlation						
Variables		CI	Age	Seniority	Working Hours	Number of Working Days Lost
Age	(r)	0.116 *	1	-	-	-
	N	447	4149	-	-	-
Seniority	(r)	0.133 **	0.762 **	1	-	-
	N	447	4149	4149	-	-
Working hours	(r)	−0.046	−0.053 **	−0.045 **	1	-
	N	447	4149	4149	4149	-
Number of working days lost	(r)	0.254 **	0.107 **	0.117 **	0.032 *	1
	N	447	4149	4149	4149	4149

* The correlation is significant at the 0.05 level (2 extremities). ** The correlation is significant at the 0.01 level (2 extremities).

## Data Availability

The data and information in this article can be found in the doctoral thesis: bsenteísmo-doença de servidores públicos do poder legislativo no Brasil, available from 6 December 2023, in the respository of the University of Porto, Portugal, at the request of the author who decided to publish a book, based on his research. access link: https://repositorio-aberto.up.pt/handle/10216/147784.

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
