# Peer review of "Prevalence of Comorbidity and Its Effects on Sickness-Absenteeism among Brazilian Legislative Civil Servants"

_ijerph, 2023, doi:10.3390/ijerph20065036_

Round 1
Reviewer 1 Report
The authors present the results of an evaluation of the impact of comorbidity on absenteeism and days not worked in the described study population. This is an important topic - if such a study is methodically well done, it makes an important contribution to recent research.
However, significant mistakes are already apparent on the first few pages. Along with this, language barriers make it very difficult to carry out a detailed review to assess the quality of the study.Please contact the editor for further information.
Author Response
I apologize for the delay in responding to observations/comments.
The article has been fully revised, including the English translation, in accordance with the reviewer's notes. Please see details in attachment.
Thank you for the corrections pointed out.

Reviewer 2 Report
The work addresses a topic that is unfortunately still current and at the same time painful for many people, included legislative public workers from all over the world.
The paper is one that is sure to attract the attention of the scientific community, and not only it. The authors are exploring a place that generates useful ideas, especially regarding health policy, which is complementary and essential for other national policies and strategies such as labour market policies. Also, it also offers important conclusions at the level of professional organizations, for decision-makers regarding the management of health and safety at work, ensuring well-being at work, with an impact on work productivity.
The issues that are addressed in the working paper are relevant to Brasil , and not only in achieving the goals of improving the health and safety of workers at work; harmonization of working environment conditions, even the public health policies and the education of the population regarding the prevention of illness.
The structure of the manuscript is clear. The work follows a well-known methodologies and theories, with innovation by mixes/ compositions of them in order to obtain the results, and it is original by applying the questionnaires and the calculation of the comorbidity index to the specific case of Brazilian legislative public workers.
The quality of the writing is generally good, but it is necessary to verify the translation of the entire text in English (see lines: 257, 267,299, and the entire table with no. 7- in text,line 320).
Attention is recommended and the renumbering of both, figures and tables:
- the first figure, line 93, is called figure 16, and on line 267 figures 1 and 2 appear;
- Tabela 5 - table 5 row 225 and 226, although previously there were only 2 tables, and then it continues with the faulty numbering;
- on line 295, a figure 30 is mentioned, but does not exist in the work.
Suggestion: since it has been mentioned that the paper is part of a larger project, a sentence about that project could make a better connection with the purpose of the present paper and would draw the reader's attention to the authors' work.
Congratulations to the authors for their work!
Continued success and many achievements in the new year, 2023!
Author Response
I apologize for the delay in responding to observations/comments.
The article has been fully revised, including the English translation, in accordance with the reviewer's notes. Please see details in attachment.
